REVIEW-SYMPOSIUM

# The role of maternal choline, folate and one-carbon metabolism in mediating the impact of prenatal alcohol exposure on placental and fetal development

Sarah E. Steane [ID], James S. M. Cuffe [ID] and Karen M. Moritz [ID]

*School of Biomedical Sciences, The University of Queensland, St Lucia, QLD, Australia*

Handling Editors: Laura Bennet & Janna Morrison

The peer review history is available in the Supporting information section of this article (https://doi.org/10.1113/JP283556#support-information-section).

This symposium review forms part of the 'Maternal influences on fetal development' symposium held at Australian Physiological Society scientific meeting on 22-25 November 2020, and organised by Dr Deanne Hryciw.

The Journal of Physiology

**Abstract**  Prenatal alcohol consumption (PAE) may be associated with a broad spectrum of impacts, ranging from no overt effects, to miscarriage, fetal growth restriction and fetal alcohol spectrum disorder. A major mechanism underlying the effects of PAE is considered to be altered DNA methylation and gene expression. Maternal nutritional status may be an important factor in determining the extent to which PAE impacts pregnancy outcomes, particularly the dietary micronutrients folate and choline because they provide methyl groups for DNA methylation via one carbon metabolism. This review summarises the roles of folate and choline in development of the blastocyst, the placenta and the fetal brain, and examines the evidence that maternal intake of these micronutrients can modify the effects of PAE on development. Studies of folate or choline deficiency have found reduced blastocyst development and implantation, reduced placental invasion, vascularisation and nutrient transport capability, impaired fetal brain development, and abnormal neurodevelopmental outcomes. PAE has been shown to reduce absorption and/or metabolism of folate and choline and to produce similar outcomes to maternal choline/folate deficiency. A few studies have demonstrated that the effects of PAE on brain development can be ameliorated by folate or choline supplementation; however, there is very limited evidence on the effects of supplementation in early pregnancy on the blastocyst and placenta. Further studies are required to support these findings and to determine optimal supplementation parameters.

(Received 4 July 2022; accepted after revision 30 January 2023; first published online 8 February 2023)

**Corresponding author** K. Moritz: School of Biomedical Sciences, The University of Queensland, Sir William MacGregor Building, St Lucia, QLD 4072, Australia.    Email: k.moritz1@uq.edu.au

**Abstract figure legend** Summary of the proposed mechanism through which maternal nutrition and one carbon metabolism (1CM) mediate the impacts of prenatal alcohol exposure (PAE). Alcohol consumption is associated with poor nutrient intake and altered absorption and/or metabolism of many nutrients. Folate and choline are micronutrients with several important roles in fetal development including epigenetic programming. Both folate and choline contribute to the transfer of methyl groups for the remethylation of homocysteine (HCY) to methionine (MET), which is used to generate the universal methyl donor *S*-adenosylmethionine (SAM) during one carbon metabolism (1CM). SAM is the substrate for the methylation of DNA (DNAm) (indicated by red circles on the DNA molecule), a key mechanism in the epigenetic regulation of gene expression. DNAm is important for epigenetic reprogramming of the blastocyst and for optimal placental and fetal development. Prenatal alcohol exposure alters DNAm, possibly by altering the availability of folate and choline, thereby disturbing 1CM.

## Introduction

Clinical studies have found that prenatal alcohol exposure (PAE) increases the risk of adverse pregnancy outcomes including early miscarriage, stillbirth, preterm birth and low birthweight (Aliyu et al., 2008; Cornman-Homonoff et al., 2012; Kesmodel et al., 2002; Patra et al., 2011). PAE also increases the risk of fetal alcohol spectrum disorder (FASD), a group of lifelong physical and neuro-logical abnormalities (Popova et al., 2016). FASD is the most common preventable neurodevelopmental disorder worldwide (CDC, 2020) with a prevalence estimated to range from 1% to 5% in the USA and Europe (May et al., 2018), and over 11% in South Africa (Roozen et al., 2016). In Australia, although the nationwide prevalence of FASD is not known, studies conducted in a remote indigenous community in Western Australia reported a prevalence of 19.4% (Fitzpatrick et al., 2017).

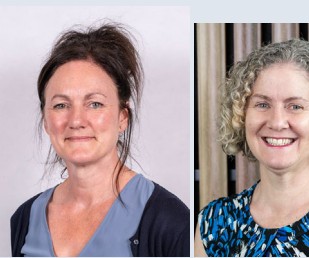

Sarah Steane is a PhD candidate at the University of Queensland, funded by a National Health and Medical Research Council Postgraduate Scholarship. Her research focuses on the effects of prenatal alcohol exposure on maternal, placental and fetal micronutrients, with a particular focus on folate and choline because of their roles in one carbon metabolism. **Karen Moritz** leads a research team at the University of Queensland that has identified the long-term impacts of alcohol during early pregnancy. She demonstrated in a rodent model that alcohol around the time of conception resulted in fetal growth restriction, insulin resistance, altered renal function, and changes to the brain and behaviour. Her team has identified that these changes are partly the result of the effects of alcohol on the pre-implantation embryo impacting early placental development.

In many countries, guidelines recommend that women abstain from alcohol when planning and during pregnancy (CDC, 2021; NHMRC, 2020; NICE, 2021) on the basis that there is no known safe amount or timing of alcohol during pregnancy. However, around 80% of women in many Western countries consume alcohol during pregnancy, typically prior to pregnancy recognition, after which most women report ceasing or reducing alcohol consumption (O'Keeffe et al., 2015). The amount of alcohol consumed and the gestational timing of consumption are known to be important risk factors for fetal outcomes, with binge level consumption (four or more standard drinks on one occasion) around conception carrying the highest risk (Ernhart et al., 1987; Kesmodel et al., 2019). These first days of pregnancy are a particularly sensitive time in development as the blastocyst undergoes epigenetic reprogramming. This involves removal of methyl groups from DNA derived from the gametes to restore totipotency to the blastocyst cells, which is followed by the establishment of new DNA methylation (DNAm) patterns that direct gene expression for placental and fetal development. Studies using animal models of alcohol exposure in early pregnancy have found altered DNAm and gene expression in the placenta and fetus in late gestation, as well as metabolic and neuro-behavioral abnormalities in adult offspring (Burgess et al., 2019; Gårdebjer et al., 2014, 2015). Such changes have been proposed to be a primary mechanism in FASD (Gutherz et al., 2022). However, it is not clear why some pregnancies exposed to alcohol result in infants with serious alcohol-related conditions, whereas other similarly exposed pregnancies produce infants with no detectable effects (Abel, 1995). This is important with respect to understanding how to develop strategies that reduce the impact of PAE.

One factor that probably contributes to the diverse spectrum of effects following PAE in women is maternal nutritional status, which may confer protection against, or susceptibility to, the effects of alcohol. Many women do not achieve the recommended intake of nutrients during pregnancy (May et al., 2016) and deficiencies in any one of the many essential micronutrients can impair placental function and lead to adverse outcomes (Richard et al., 2017). Nutritional deficiencies are probably exacerbated by alcohol consumption, which is associated with both poor nutrient intake and altered absorption of many nutrients (Bode & Bode, 2003; Kloss et al., 2022; Sebastiani et al., 2018). The impact of PAE on the micronutrients folate and choline is of particular interest because of their overlapping roles as methyl donors for DNAm, as well as their distinct roles in placental and fetal development. This review will summarise current knowledge on the roles of folate and choline on development of the blastocyst, placenta and fetal brain, and examine the evidence for an interaction between these micronutrients and PAE in determining pregnancy outcomes.

## The importance of folate and choline during pregnancy

Folate is an essential nutrient that must be obtained from dietary sources such as green leafy vegetables. The importance of periconceptional folate for prevention of fetal neural tube defects is well understood and led to fortification of food products with folic acid, in over 80 countries, and inclusion in prenatal vitamins. The recommended daily intake of folate when planning a pregnancy and during the first trimester is 600–800 $\mu$g day$^{-1}$ (IOM, 1998; National Health & Medical Research Council, 2006); however, many women planning a pregnancy do not achieve this (McDougall et al., 2021). The recommended folate intake increases to 5000 $\mu$g day$^{-1}$ in women with risk factors for neural tube defects, such as obesity (RANZCOG, 2019). Experiments in mice suggest that folate is important from the very early stages of development. Preimplantation embryos removed from pregnant mice (two- and eight-cell stages) have been found to express folate receptor 1 (FR1), which mediates folate uptake by endocytosis and the reduced folate carrier (RFC) antiporter, (eight-cell stage only) (Kooistra et al., 2013). The same study also demonstrated uptake of [$^3$H] folate into preimplantation embryos, which was inhibited by blocking endocytosis indicating transport via FR1. In the placenta, FR1 and RFC, together with the proton-coupled folate transporter transfer folate across the synctiotrophoblast of the placenta into the fetal circulation to a level two to four times higher than in the maternal circulation (Giugliani et al., 1985; Radziejewska & Chmurzynska, 2019). Folate supplies methyl groups for both the generation of nucleotides for DNA replication during cell division, and for DNAm.

Choline can be synthesised in the liver; however, endogenous production is insufficient to meet physiological demands. Therefore, additional choline must be obtained from dietary sources, including eggs, meat and dairy products. The adequate intake (AI) of choline is 425 mg day$^{-1}$ for non-pregnant women and 450 mg day$^{-1}$ for pregnant women (IOM, 1998). Similar to folate, choline is also important for closure of the neural tube (Shaw et al., 2009); however, unlike folate, choline is not fortified in foods and is not a standard component of prenatal vitamins. Fewer than 10% of pregnant women in the US are considered to obtain sufficient choline from the diet (Wallace & Fulgoni, 2017), which is of concern given that the requirement for choline increases dramatically during pregnancy and the AI of 450 mg day$^{-1}$ is considered to underestimate requirements (Yan et al.,

2013). Similar to folate, uptake of choline by one- and two-cell conceptuses and preimplantation blastocysts flushed from the mouse oviduct and uterus, respectively, has been demonstrated *in vitro* by addition of [³H] choline to the culture medium (Van Winkle et al., 1993). Choline transport activity was over 100 times greater in the blastocyst compared with the two-cell conceptus, indicating a role in blastocyst development. Choline is actively transported across the placenta, predominantly by the choline transporter-like proteins 1 and 2 (CTL1/2). CTL1 protein has been detected in the human placenta, localised to sycytiotrophoblasts of the chorionic villi, from 6 to 8 weeks of gestation (earliest time point examined), with expression remaining consistent throughout pregnancy (Baumgartner et al., 2015). Toward the end of the first trimester, CTL1 was also found in the endothelium of the fetal vasculature. CTL2 expression was detected in the stroma of chorionic villi in early gestation and was also co-localised with CTL1 in the fetal vasculature around 18 weeks. Choline is present in cord blood at a concentration three- to five-fold higher than in the maternal circulation at birth (Zeisel et al., 1980). Choline has several significant functions in fetal development, being the precursor of membrane components such as phosphatidylcholine and sphingomyelin, and the neurotransmitter ACh, important in fetal brain and placental development (Korsmo et al., 2019). Similar to folate, choline is also a methyl donor for DNAm, which is central to the epigenetic regulation of gene expression.

## Folate and choline metabolism are inter-related by one carbon metabolism (1CM)

In the liver, both folate and choline (via betaine), provide methyl groups for the remethylation of homocysteine to methionine, which is used to generate the universal methyl donor S-adenosylmethionine (SAM) during 1CM (Figure 1). SAM provides methyl groups for numerous methyltransferase-dependent pathways including DNAm, which is catalysed by DNA methyltransferases (DNMTs). Folate and choline are metabolically linked by their shared role as methyl donors in 1CM such that dietary deficiency of either one results in a secondary deficiency of the other as a result of the compensatory provision of methyl groups for 1CM (Kim et al., 1994; Selhub et al., 1991). Deficiency in either choline or folate results in reduced methionine and SAM concentrations in the rat liver (Zeisel et al., 1989). Hepatocytes are one of the few cell types able to use methyl groups from the choline pathway, with most cells being solely dependent on folate-derived methyl groups. However, both folate and choline derived methyl groups also appear to be important in early blastocyst development, which is the period of gestation considered to be most vulnerable to the effects of alcohol.

## The impact of alcohol on absorption and metabolism of folate and choline

Alcohol consumption inhibits intestinal folate absorption (Halsted et al., 2002), with chronic alcohol consumption

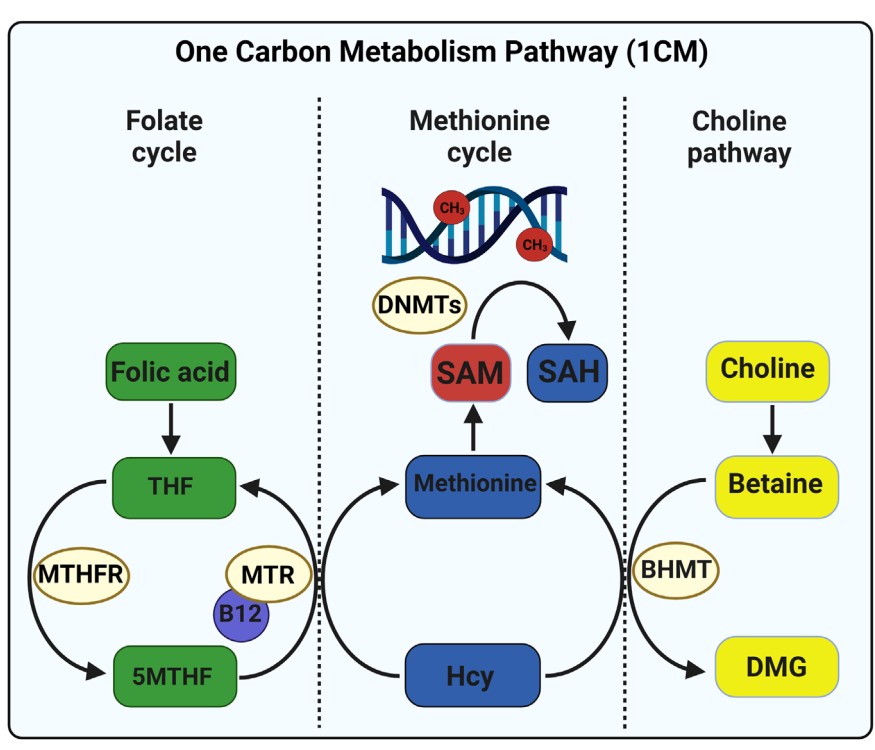

**Figure 1. The 1CM cycle and the inter-relationship between folate and choline as methyl donors for DNA methylation**
The 1CM cycle demonstrating the inter-relationship between folate and choline and their contribution to the regulation of DNA methylation (adapted with permission from Steane et al., 2022). The 1CM pathway is divided into the *folate cycle*, *methionine cycle* and the *choline pathway*. The red circles on the DNA molecule represent methyl groups (CH$_3$). THF, tetrahydrofolate; 5MTHF, 5-methyl-tetrahydrofolate; MTHFR, methylenetetrahydrofolate reductase; MTR, methionine synthase; DNMTs, DNA methyltransferases; SAM, S-adenosylmethionine; SAH, S-adenosylhomocysteine; Hcy, homocysteine; DMG, dimethylglycine; BHMT, betaine homocysteine methyltransferase. A recent review of 1CM is provided in Lionaki et al. (2022).

reducing plasma folate levels by as much as 80% (Halsted et al., 1971). Alcohol also inhibits activity of methionine synthase, the enzyme that transfers a methyl group from folate to homocysteine, thereby reducing the ability to use folate as a methyl donor for 1CM (Barak et al., 2002) (Figure 1). Although there is currently no evidence to suggest that alcohol directly reduces absorption of dietary choline, it is assumed to increase the demand for choline in at least two ways. First, choline is required for the export of hepatic lipids generated by alcohol metabolism via formation of very low density lipoproteins (Cole et al., 2012). Second, alcohol induced folate deficiency probably results in a secondary deficiency in choline as a compensatory mechanism to provide methyl groups for 1CM. This is supported by the finding that alcohol feeding in rats reduces hepatic methionine and SAM and reduces DNAm by 40% (Lu et al., 2000). Taken together, altered absorption and/or demand for folate and choline as a result of alcohol consumption probably exacerbates the inadequate dietary intake and increased requirements for these factors during pregnancy. This may be particularly important during early pregnancy because the developing blastocyst is highly sensitive to changes in micronutrient availability (Dos Santos Mendonça-Soares et al., 2022; Estrada-Cortés et al., 2020; Menezo et al., 2022).

### Roles of folate and choline in blastocyst development

Following fertilisation of the oocyte, the resulting zygote undergoes cell divisions to form the blastocyst, which differentiates into an outer trophectoderm layer that will form the placenta, and an inner cell mass that will form the fetal tissues. Folate is critical for early embryonic development, providing methyl groups for the synthesis of nucleotides required for DNA replication, prior to cell division. This is underscored by the clinical use of the folate antagonist methotrexate to disrupt embryo development in ectopic pregnancies (extrauterine blastocyst implantation) (Shiravani et al., 2021). However, the importance of folate in early blastocyst development extends beyond nucleotide synthesis, with folate derived methyl groups being required for epigenetic reprogramming of the blastocyst. This reprogramming event refers to the removal of gamete specific DNAm patterns that direct gamete specific gene expression, followed by the establishment of new DNAm patterns that direct expression of genes required for fetal and placental development. Experiments using mouse preimplantation blastocysts have demonstrated that both folate and choline are important contributors of methyl groups for DNAm in the blastocyst (Zhang et al., 2015). When blastocysts were treated with methotrexate to block the folate pathway but supplemented with nucleotides to allow normal DNA synthesis, immunofluorescence staining of methylated

cytosine residues in DNA (5-MeC) was significantly reduced. A similar outcome was observed when the choline pathway was disrupted by knocking down betaine homocysteine methyltransferase (BHMT), whereas disruption of both pathways simultaneously almost completely abolished 5-MeC. Interestingly, blocking either pathway individually reduced cell number in the inner cell mass, whereas, in the trophectoderm, cell number was only reduced when the folate pathway was disrupted. This suggests that SAM is important for cell proliferation and that, although the inner cell mass utilises both folate and choline for generation of SAM, the cells of the trophectoderm depend entirely on folate. Blocking both pathways also reduced implantation rates in embryo transfer experiments, whereas blocking each pathway individually had no effect. However, there were more resorptions in the methotrexate treated blastocysts, perhaps providing further support for the importance of folate in placental development. Dose-dependent alterations in blastocyst DNAm and gene expression have been demonstrated for both folate (Rahimi et al., 2019) and choline (Estrada-Cortés et al., 2020).

### Interaction between PAE, folate and choline on blastocyst development

Ethanol exposure has also been shown to alter DNAm in the preimplantation blastocyst. Exposure of rats to ethanol around the time of conception and up until blastocyst implantation resulted in hypermethylation of the blastocyst, a reduction in maternal plasma choline and an increased number of resorptions (Kalisch-Smith et al., 2019). Similarly, a study using mouse blastocysts demonstrated hypermethylation following ethanol exposure *in vitro*, a decreased number of implantations following embryo transfer and fetal growth restriction (Zhang et al., 2018). However, concurrent supplementation of the blastocyst culture with betaine normalised DNAm and implantation rates and reduced the proportion of fetuses that were growth restricted. This may indicate that alcohol reduces transport of folate into the blastocyst and, although betaine is able to compensate methyl groups in the inner cell mass and normalise implantation rates, trophectoderm cells are unable to use betaine as a methyl donor, resulting in poor placentation and subsequent fetal growth restriction. Further studies are required to investigate whether supplementation with folate, or both folate and choline *in vivo*, can further ameliorate the effects of alcohol on the blastocyst and restore placental and fetal development.

### Roles of folate and choline in placental development

Low folate status in women in the third trimester of pregnancy has been found in association with reduced

birthweight and placental abnormalities including apoptosis of trophoblast cells, decreased activity of amino acid transporters, increased expression of miRNAs and altered gene expression (Baker et al., 2017; Rosario et al., 2017). Folate supplementation prior to and during the first 12 weeks of pregnancy at 4 mg day$^{-1}$ compared to 0.4 mg day$^{-1}$ in women resulted in increased birthweight and a reduced incidence of spontaneous abortion, small for gestational age and preterm birth, hypothesised to be a result of improved placental development (Bortolus et al., 2021). Studies using cultured human placental cell lines demonstrated that low folate in the culture medium reduced trophoblast viability and invasive capability, and was also associated with altered DNAm and reduced expression of growth factors and matrix metalloproteases, which are important in the invasion process (Ahmed et al., 2016; Rahat et al., 2022). A folate-deficient diet throughout pregnancy in mice also resulted in poor trophoblast invasion, altered placental morphology and placental abruption (Pickell et al., 2009). These studies indicate that folate deficiency results in poor placentation, possibly as a result of reduced availability of methyl groups, which may limit cell proliferation by reducing nucleotide synthesis, as well as impacting cell function (e.g. invasive capability) by altering DNAm and therefore gene expression patterns (Figure 2).

Similar to folate, choline supplementation in women has also been demonstrated to alter placental DNAm and gene expression. However, unlike folate, choline does not directly provide methyl groups in the placenta because BHMT is not expressed (Solanky et al., 2010). Choline supplementation probably provides methyl groups indirectly via maternal hepatic 1CM metabolism and/or by sparing folate for placental 1CM (Steane et al., 2022). Third trimester choline supplementation in women of 930 mg day$^{-1}$ compared to 480 mg day$^{-1}$, which is approximately the adequate intake level, resulted in a 22% increase in placental global DNAm (Jiang et al., 2012). Placental expression of 43 genes were upregulated and 123 were downregulated in the 930 mg day$^{-1}$ vs. 480 mg day$^{-1}$ group, many of which have roles in placental vascular function (Jiang et al., 2013). A larger cohort study found that maternal plasma choline concentration in the first and third trimester negatively correlated with global (LINE-1) and gene specific DNAm in the placenta (Nakanishi et al., 2021). Conflicting results on the relationship between placental global DNAm and maternal choline intake *vs.* plasma concentration raise the question of how well measurements in maternal plasma reflect maternal nutrient intake and distribution to the placental and fetal compartments (Mujica et al., 2020).

Preclinical studies of maternal choline supplementation have also produced somewhat conflicting results. Choline supplementation in pregnant mice was found to modify markers of inflammation, apoptosis and vascularisation in the placenta and to increase the luminal area of maternal spiral arteries (Kwan et al., 2017), and a follow-up study found increased placental global DNAm at embryonic day (E)15.5 and altered methylation at the promoters of several imprinted genes (Kwan et al., 2018). Conversely, choline supplementation in pregnant rats resulted in decreased placental global DNAm, increased insulin-like growth factor 2 (*Igf2*) expression and increased placental efficiency at E20 (Steane, Fielding et al., 2021). The different effects of choline on DNAm in these studies may be a result of the different gestational time points examined, an area that requires further investigation.

## Interaction between PAE, folate and choline on placental development

A recent systematic review and meta-analysis of placental outcomes following PAE in women found that PAE increases the likelihood of placental abruption and of reduced placental weight (Steane, Young et al., 2021). Included studies also provided evidence for altered placental DNAm, gene expression and altered placental vasculature, indicative of maternal vascular malperfusion. Preclinical studies have shown that alcohol exposure around the time of blastocyst implantation results in poor trophoblast invasion and incomplete remodelling of uterine arteries (Gundogan et al., 2008; Kalisch-Smith et al., 2019). These effects may be mediated by the effect of alcohol on placental folate transport (Figure 2) because alcohol has been shown to reduce the expression of folate transporters and uptake of labelled folic acid in human placental trophoblast cells (Keating et al., 2008; Keating et al., 2009). Placental transfer of folate from maternal to fetal circulation has also been shown to be reduced in women with PAE (Hutson et al., 2012).

Preclinical studies have provided some evidence that low maternal folate exacerbates the effects of alcohol on pregnancy outcome. In pregnant mice fed a folate deficient diet and given either a low or high dose of alcohol on E11, both doses resulted in decreased placental weight, fetal abnormalities and fetal death, whereas, when mice were fed a folate replete diet, only the high dose of ethanol produced these detrimental effects (Gutierrez et al., 2007). Similarly, in pregnant rats fed a folate free diet, acute ethanol exposure at E11 resulted in a significant increase in resorptions and a reduction in viable fetuses, whereas there were no effects of PAE in dams receiving a folate sufficient diet (Lin, 1988).

Prenatal choline has also been shown to reduce the effects of PAE in preclinical studies. Choline supplementation in a rat model of periconceptional ethanol exposure reduced placental global DNAm, and increased expression of the growth factor *Igf2*, as well as the proportion of labyrinth zone (region of nutrient transfer), resulting in increased placental efficiency and

fetal growth in late gestation (Steane, Fielding et al., 2021). Although these positive outcomes were observed in both the control and ethanol groups, the magnitude of effects was greater in ethanol-exposed animals. In a follow-up study, ethanol was found to reduce several components of 1CM in the placenta (including choline, folate and the SAM:SAH ratio), whereas choline supplementation restored levels to those seen in non-exposed control animals receiving a standard diet (Steane et al., 2022).

Similarly, a study of alcohol exposure in mice from E8.5 to E17.5 found that concurrent choline supplementation increased placental efficiency, prevented reductions in fetal body weight and normalised indicators of brain sparing (Kwan et al., 2021).

Overall, these studies provide some evidence that the effects of PAE, particularly in early pregnancy, may be mediated at least in part by its effects on folate transport and metabolism. Folate supplementation may directly

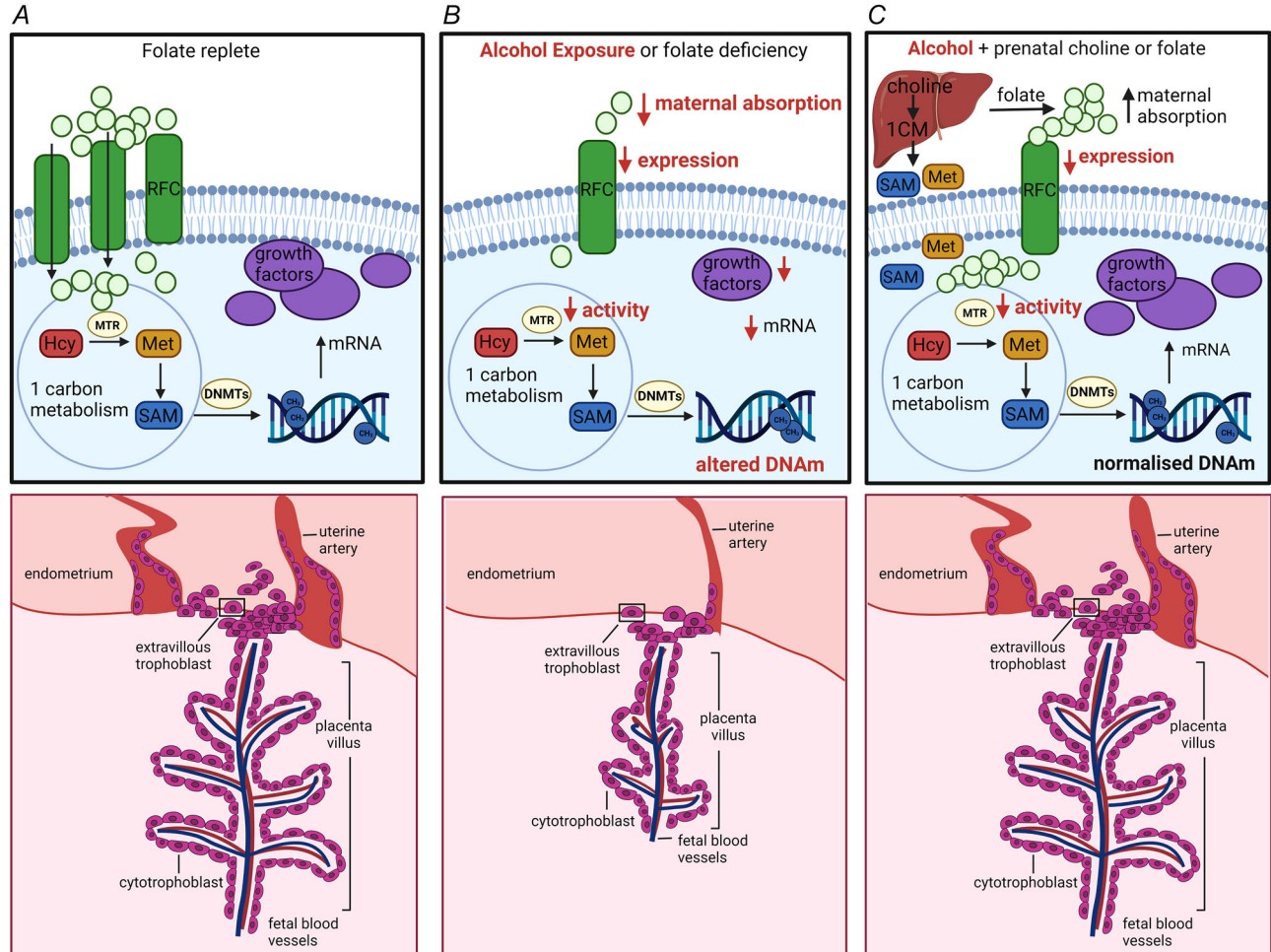

**Figure 2. Proposed mechanism for the effects of prenatal alcohol exposure and methyl donors on early placental development**

Proposed molecular changes in trophoblast cells (upper) and consequences for placental invasion and development (lower) in relation to (*A*) folate replete maternal environment, (*B*) maternal alcohol exposure or folate deficiency and (*C*) maternal alcohol exposure and supplementation with folate or choline. Folate is transported into the trophoblast cell via the reduced folate carrier (RFC) (Radziejewska & Chmurzynska, 2019). Methionine synthase (MTR) catalyses the transfer of a methyl group from folate for the remethylation of homocysteine (Hcy) to methionine (Met), which is then used to generate *S*-adenosylmethionine (SAM). DNA methyltransferases (DNMTs) catalyse the transfer of methyl groups from SAM to DNA establishing optimal DNAm for the transcription of growth factors such as vascular endothelial growth factor (Vegfa) and other proteins involved in trophoblast invasion such as matrix metalloproteinases (Rahat et al., 2022). PAE reduces expression of RFC (Keating et al., 2009) and activity of MTR (Barak et al., 2002), which may alter DNAm and transcription of growth factors, thereby impacting on trophoblast invasion and placentation (Gualdoni et al., 2021; Williams et al., 2011). Supplementation with choline probably provides alternate methyl donors in the maternal liver, sparing folate and producing additional SAM and methionine for delivery to the placenta (Steane et al., 2022), normalising DNAm, gene transcription and placentation.

overcome the effect of alcohol on folate metabolism in the placenta, whereas choline supplementation probably improves outcomes by compensatory provision of methyl groups in the maternal liver, sparing folate for placental and fetal development (Figure 2). Given that the placenta plays a critical role in fetal development by supplying nutrients and by the secretion of growth hormones and neurotransmitters, any disruptions to the placenta may have a direct impact on fetal brain development (Santos et al., 2020).

## Roles of folate and choline in fetal brain development

Clinical studies have reported that children of women who did not take folic acid supplements in the first trimester of pregnancy were found to have poorer executive control, motor ability and receptive language ability at 4 years of age compared to children of mothers who supplemented with folate (Neumann et al., 2019). Folate intake in the first trimester has also been found to be positively associated with child cognition at age 3 years (Villamor et al., 2012). Folic acid supplementation throughout pregnancy is associated with improved neurocognitive performance in children up to age 11 years (Caffrey et al., 2021) and altered DNAm at promoters of brain related genes in cord blood from the same children (Ondičová et al., 2022). There is also evidence for a long-term effect of maternal folate supplementation on DNAm in adulthood (Richmond et al., 2018). A recent systematic review concluded that prenatal folic acid supplementation had a positive impact on child measures of neurodevelopment and reduced the risk of behavioural and language abnormalities (Chen et al., 2021).

Preclinical studies have further indicated a mechanistic link between maternal folic acid intake, DNAm in the fetal brain and neurodevelopmental outcomes. Maternal folate supplementation in rats at a level equivalent to 400 $\mu$g day$^{-1}$ in women, increased neurogenesis and synaptogenesis in the neonatal brain (Wang et al., 2019), whereas maternal folate deficiency in rats resulted in decreased proliferation and increased apoptosis in neural progenitor cells in the fetal brain (Craciunescu et al., 2004). Further studies demonstrated a positive association between prenatal folate and measures of sensory motor function, memory and learning in adolescent and adult offspring (Wang et al., 2018). Follow-up studies demonstrated improvements in sensory motor function to be associated with an increased SAM:SAH ratio, DNAm, and the expression and activity of DNMT1, DNMT3a and DNMT3b (Li et al., 2018). However, recent studies have demonstrated that feeding pregnant mice a diet containing five-fold the standard amount of folate resulted in altered gene expression in the placenta and brain of offspring, which was associated with hyperactivity and impaired memory (Luan et al., 2022).

Studies of prenatal choline supplementation in women have provided mixed results regarding cognitive outcomes in children. The same randomised control trial that examined choline supplementation and placental DNAm, as described above (Jiang et al., 2012), also found that prenatal supplementation with 930 mg day$^{-1}$ choline compared to 480 mg day$^{-1}$ was associated with increased infant information processing speed (Caudill et al., 2018). Prenatal choline intake from dietary sources has also been found to be positively associated with visual memory in children at age 7 years (Boeke et al., 2013). Maternal plasma choline concentration has been found to be positively associated with infant cognition in one study (Wu et al., 2012), whereas, in another, there was no association between either maternal or cord blood choline concentration and measures of IQ and memory in children at age 5 years (Signore et al., 2008). However, preclinical studies have demonstrated that maternal choline supplementation in rats improved memory in offspring (Meck et al., 1988), whereas maternal choline deficiency in mice reduced neurogenesis and angiogenesis in the hippocampus of offspring and increased expression of vascular endothelial growth factor and angiopoietin 2 (Craciunescu et al., 2003; Mehedint et al., 2010).

Some studies have investigated the interaction between choline and folate on brain development. A study in mice found that a folate deficient diet in late gestation decreased proliferation and increased apoptosis in neural progenitor cells in a number of brain regions and concurrent choline supplementation was able to restore some, but not all, of these deficits. (Craciunescu et al., 2010). Similarly in mice carrying a mutation that disturbs folate metabolism, mild choline deficiency altered maternal 1CM and increased the occurrence of developmental defects (Jadavji et al., 2015). A study examining combinations of low, standard and high dietary folate and choline in pregnant mice found differential effects on offspring 1CM and hypothalamic gene expression, which were associated with altered food intake (Hammoud et al., 2021). Given the inter-related roles of choline and folate as methyl donors, and the impact on fetal programming and offspring outcomes, further studies are required to determine the optimal balance between maternal intake of folate and choline for offspring health. This is particularly important when absorption of micronutrients may be compromised, as is the case with alcohol consumption, as discussed below.

## Interaction between PAE, folate and choline on fetal brain development

Interactions between PAE and maternal dietary folate and their impact on brain development have been examined in animal models (Figure 3). In pregnant rats exposed to alcohol from E6 onward, folic acid supplementation throughout gestation reduced PAE-induced neuro-

apoptosis in the brain of neonates (Sogut et al., 2017). Concurrent supplementation of alcohol exposed mice with folic acid (E6 onward) reduced the effects of PAE on motor activity and anxiety (Shrestha & Singh, 2013). Using a mouse model of PAE (from E6 to E15), it has been demonstrated that concurrent supplementation with folate ameliorated expression changes in the brain of offspring for 17 of 30 proteins that were altered by alcohol (Xu et al., 2008). Supplementation also normalised alcohol-induced reductions in fetal brain weight, bodyweight and brain:body weight ratio. However, in a guinea-pig model, chronic ethanol exposure reduced fetal body and brain weight, and this was not mitigated by folic acid supplementation, which may be a result of the lower dose of folic acid in this study (Hewitt et al., 2011).

Prenatal choline supplementation in women with PAE has been found to have positive effects on infant development (Figure 3). A trial in pregnant women with heavy alcohol use found that supplementation with 2 g day$^{-1}$ choline, beginning from recruitment (prior to 23 weeks of gestation), improved infant eye blink conditioning, a response found to be absent or impaired in FASD and heavily PAE exposed infants, and improved scores on tests of visual recognition memory at 12 months (Jacobson et al., 2018). A follow-up study in the same cohort found that choline supplementation mitigated PAE-induced reductions in infant brain volume and was associated with improved memory (Warton et al., 2021). PAE has also been shown to reduce scores on infant psychomotor and cognitive development

measures at 6 months, whereas supplementation with a multivitamin improved outcomes on cognitive measures (Coles et al., 2015). However, daily supplementation with multivitamins plus choline (750 mg) had no additional benefit. A recent cohort study found that, in women with PAE, reduced gestational weight gain and low intakes of energy, choline and iron were associated with more severe fetal growth restriction (Carter et al., 2022).

Animal models of PAE have found that prenatal choline could ameliorate the impact of alcohol on neuro-behavioural outcomes (Idrus et al., 2017; Thomas et al., 2010) and have provided insights into the underlying mechanisms. In mice fed a liquid diet containing 25% ethanol either with or without choline chloride from E0.5 until birth, PAE resulted in DNA hypomethylation and altered gene expression in the neocortex, both of which were prevented by choline (Bottom et al., 2020). A rat model of PAE from E7 onward found increased expression of DNMT1, increased methylation of the hypothalamic proopiomelanocortin (POMC) gene and reduced expression of POMC, all of which were normalised by concurrent choline supplementation (Bekdash et al., 2013).

Taken together, these studies indicate that prenatal supplementation with either folate or choline can ameliorate some of the detrimental effects of alcohol on the fetal brain. This is probably at least partly a result of the positive effects of supplementation on preventing or reversing perturbed methylation of genes that impact placental function (Figure 3).

## Conclusions

Clinical and preclinical studies demonstrate that maternal deficiency of either folate or choline alters DNAm and gene expression in the developing blastocyst, as well as both the placenta and fetal brain. Many women do not achieve the recommended daily intake of folate and many more do not meet the adequate intake for choline, an intake level that is now considered to underestimate the increased requirements for choline during pregnancy. These women may be particularly vulnerable to the additional effects of alcohol on folate and choline metabolism. The combined effects of inadequate folate/choline and PAE have been demonstrated to result in the most detrimental effects on the fetus by impacting both the placenta and the fetal brain. Outcomes are dependent at least in part on the timing of PAE, which may be particularly harmful around conception, a time of high demand for methyl groups for epigenetic reprogramming of the blastocyst. Changes in DNAm in the blastocyst are associated with reduced invasion of placental trophoblast cells, which may lead to incomplete spiral artery remodelling and reduced maternal blood supply to the developing placenta and fetus. This may

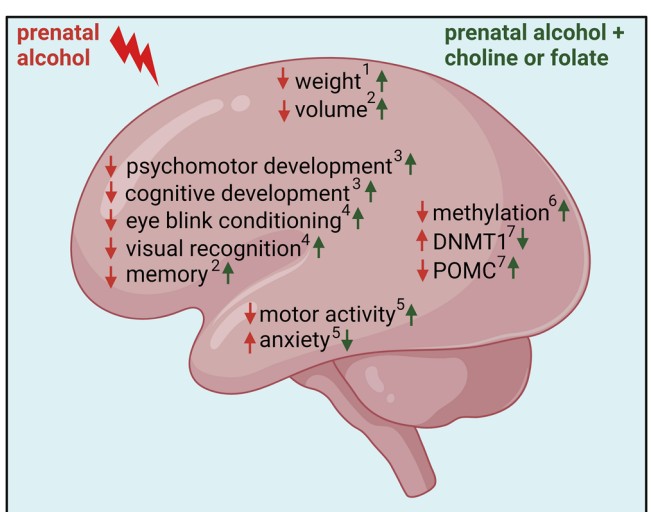

**Figure 3. Effects of alcohol exposure plus supplementation with choline or folate**
Summarising the effects of prenatal alcohol exposure (red) or prenatal exposure plus supplementation with choline or folate (green) on structural and functional development of the fetal brain: [1] (Xu et al., 2008), [2] (Warton et al., 2021), [3] (Carter et al., 2022), [4] (Jacobson et al., 2018), [5] (Shrestha & Singh, 2013), [6] (Bottom et al., 2020), [7] (Bekdash et al., 2013).

result in alterations in apoptosis and vascularisation in the developing placenta and brain, with effects on neurodevelopmental outcomes.

Further research is required to investigate whether maternal supplementation with either choline or folate, or a combination of the two, in the periconceptional period and throughout pregnancy can protect against the effects of PAE. However, given the high prevalence of alcohol consumption in women around the time of conception and the low intake of choline amongst pregnant women, it is imperative that public health messaging continues to raise awareness of the risks of PAE, as well as the importance of achieving the recommended intakes of choline and folate in the periconceptional period and continuing throughout pregnancy. Consideration could be given to adding choline as a standard component of all prenatal supplements to provide a minimum daily dose of 450 mg of choline.

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

## Additional information

### Competing interests

The authors declare that they have no competing interests.

### Author contributions

S.E.S. was responsible for the design and first draft of the article. All authors were involved with the analysis and interpretation of the data, along with drafting the article or critically revising it for important intellectual content. All authors have read and approved the final version of the manuscript submitted for publication and agree to be accountable for all aspects of

the work in ensuring that questions related to the accuracy or integrity of any part of the work are appropriately investigated and resolved. All persons designated as authors qualify for authorship and all those who qualify for authorship are listed.

## Funding

This research was funded by National Health and Medical Research Council for support of S.E.S. (grant number APP1191217) and K.M.M. (grant number APP1078164).

## Acknowledgements

We acknowledge the assistance of Dr Lisa Akison (School of Biomedical Sciences, University of Queensland, St Lucia, QLD, Australia) with the design of the figures.

Open access publishing facilitated by The University of Queensland, as part of the Wiley - The University of Queensland agreement via the Council of Australian University Librarians.

## Keywords

blastocyst, DNA methylation, fetal brain, maternal nutrition, one carbon metabolism, placenta, prenatal alcohol

## Supporting information

Additional supporting information can be found online in the Supporting Information section at the end of the HTML view of the article. Supporting information files available:

**Peer Review History**

