## [Peer Review History · The Journal of Physiology]

The role of maternal choline, folate and one-carbon metabolism in mediating the impact of prenatal alcohol exposure on placental and fetal development

Sarah E Steane, James S M Cuffe, and Karen M Moritz

DOI: 10.1113/JP283556

Corresponding author(s): Karen Moritz (k.moritz1@uq.edu.au)

The following individual(s) involved in review of this submission have agreed to reveal their identity: Maureen Keller-Wood (Referee #2)

Review Timeline:

Submission Date:	04-Jul-2022
Editorial Decision:	30-Sep-2022
Revision Received:	16-Jan-2023
Accepted:	30-Jan-2023

Senior Editor: Laura Bennet

Reviewing Editor: Janna Morrison

Transaction Report:

Dear Dr Moritz,

Re: JP-SR-2022-283556 "The role of maternal nutrition and one-carbon metabolism in mediating the impacts of prenatal alcohol exposure on placental and fetal development" by Sarah E Steane, James S M Cuffe, and Karen M Moritz

Thank you for submitting your invited Review-Symposium to The Journal of Physiology. It has been assessed by a Reviewing Editor and by 2 expert referees and I am pleased to tell you that it is considered to be acceptable for publication following satisfactory revision.

The reports are copied at the end of this email. Please address all of the points and incorporate all requested revisions, or explain in your Response to Referees why a change has not been made.

NEW POLICY: In order to improve the transparency of its peer review process The Journal of Physiology publishes online as supporting information the peer review history of all articles accepted for publication. Readers will have access to decision letters, including all Editors' comments and referee reports, for each version of the manuscript and any author responses to peer review comments. Referees can decide whether or not they wish to be named on the peer review history document.

I hope you will find the comments helpful and have no difficulty in revising your manuscript within 4 weeks.

Your revised manuscript should be submitted online using the links in Author Tasks Link Not Available. This link is to the Corresponding Author's own account, if this will cause any problems when submitting the revised version please contact us.

The image files from the previous version are retained on the system. Please ensure you replace or remove any files that have been revised. Your revised submission should include:

- A Word file of the complete text (including figure legends any Tables);
- An Abstract Figure (with legend in the Article file)
- Each figure as a separate, high quality, file;
- A full Response to Referees;
- A copy of the manuscript with the changes highlighted.
- Author profile. A short biography (no more than 100 words for one author or 150 words in total for two authors) and a portrait photograph of the two leading authors on the paper. These should be uploaded, clearly labelled, with the manuscript submission. Any standard image format for the photograph is acceptable, but the resolution should be at least 300 dpi and preferably more.

- A 'Cover Art' file for consideration as the Issue's cover image;
- Appropriate Supporting Information (Video, audio or data set https://jp.msubmit.net/cgi-bin/main.plex?form_type=display_requirements#supp).

To create your 'Response to Referees' copy all the reports, including any comments from the Reviewing Editor into a Word, or similar, file and respond to each point in colour or CAPITALS and upload this when you submit your revision.

I look forward to receiving your revised submission.

If you have any queries please reply to this email and staff will be happy to assist.

Yours sincerely,

Professor Laura Bennet
Senior Editor
The Journal of Physiology
<https://jp.msubmit.net>
<http://jp.physoc.org>
The Physiological Society
Hodgkin Huxley House
30 Farringdon Lane
London, EC1R 3AW
UK
<http://www.physoc.org>
<http://journals.physoc.org>

REQUIRED ITEMS:

-Please upload separate high quality figure files via the submission form.

-Author profile(s) must be uploaded via the submission form. Authors should submit a short biography (no more than 100 words for one author or 150 words in total for two authors) and a portrait photograph of the two leading authors on the paper. These should be uploaded, clearly labelled, with the manuscript submission. Any standard image format for the photograph is acceptable, but the resolution should be at least 300 dpi and preferably more. A group photograph of all authors is also acceptable, providing the biography for the whole group does not exceed 150 words.

EDITOR COMMENTS

Reviewing Editor:

This is a well written and interesting summary of the mechanisms mediating the impact of prenatal alcohol exposure on fetal development. It is a valuable synthesis of the area. The figures are really helpful in summarising the findings.

Line 80 - This sentence refers to guidelines in countries but only cites NHMRC. Can other guidelines also be cited?

Line 99 -to understand HOW to develop.....

Line 101 and elsewhere - In most cases, that is preferred over which. When 'which' is used, it is generally preceded by a comma with the exceptions of 'in which', 'by which', etc.

Line 102 - which is fine but should be preceded by a comma.

Line 117 - led not lead?

Line 120 and elsewhere - However should be preceded by a semicolon.

REFEREE COMMENTS

Referee #1:

This a well written review discussing the potential role of folate and choline in mediating the effects of alcohol on the fetus with an emphasis on placental development. I have a few minor comments that should be addressed for the paper is considered for publication in the Journal of Physiology.

1. The title of 'the role of maternal nutrition....' is a little misleading when the paper focuses almost entirely on folate and choline. It might be better to be explicit and say the role of folate, choline and one carbon metabolism in....
2. Some mention of the metabolic effects of alcohol that may alter other elements of nutrition essential for feto-placental development would also be useful eg what effect does alcohol have on food intake, glucose and amino acid levels, other hormones and factors known to alter placental function. Perhaps as a table.
3. The paper would benefit from some discussion of placental folate and choline transport and the cellular mechanisms involved.
4. Graphical abstract and Figure 1. Identify that the circles on the DNA molecule represent methylation.
5. Figures 1, 2 and 3 legends should include the references from which the data has been drawn.

6. Figure 2B Should be titled 'Alcohol exposure and folate deficiency' otherwise it could be read as alcohol deficiency

7. The conclusions recommend a minimum choline dose but makes no mention of the need for a greater dose of folate during pregnancy. Should this be included?

Referee #2:

This is an interesting and well-written review, which summarizes work in this area and points out gaps in knowledge requiring further research on mechanisms, but also points out where clear associations suggest greater public health awareness and policy to improve outcomes.

The review makes a nice case for the effects of alcohol mediated through, and exacerbated by, reduced nutrients and effects on methylation. It is clear that there are different mechanism depending on the time in pregnancy of exposure to alcohol. I think it would be helpful to include a figure that shows the effects of alcohol/ reduced choline and folate over the course of pregnancy, ie early effects on blastocyst, later effects on placentation and early brain development, and even later effects on placental vascularity, "activity" (transporters etc) and fetal brain development/maturation.

Specific comments:

Page 6-7: Is there any information on folate or choline uptake into the blastocyst or early embryo in vivo (ie prior to placentation?)

Line 185-186: Is there a reference for 'This may be particularly important during early pregnancy as the developing blastocyst is highly sensitive to changes in micronutrient availability.'

Lines 248-249: The mechanism for the link between reduced availability of methyl groups and poor placentation is not discussed. What is the hypothesized mechanism? Is this due to reduced proliferative capacity or is it a specific epigenetic effect secondary to hypomethylation? Does folate also affect igf2 levels?

Concluding paragraph: makes a case for raising awareness about choline and folate intake in the periconception period, but the review also makes a case for the importance of both of these throughout pregnancy, and has evidence for an interaction of alcohol with either folate or choline deficiency in all stages of pregnancy.

END OF COMMENTS

Confidential Review

04-Jul-2022

Response to Reviewing Editor and Referees

PLEASE NOTE THAT LINE NUMBERS PROVIDED IN EACH RESPONSE REFER TO THE REVISED MANUSCRIPT WITH CHANGES HIGHLIGHTED

Reviewing Editor:

Line 80 - This sentence refers to guidelines in countries but only cites NHMRC. Can other guidelines also be cited?

Citations for guidelines from government agencies in the UK (NICE guidelines) and the US (CDC guidelines) have been added (line 82).

Line 99 -to understand HOW to develop.....

Corrected (line 101)

Line 101 and elsewhere - In most cases, that is preferred over which. When 'which' is used, it is generally preceded by a comma with the exceptions of 'in which', 'by which', etc.

Corrected (highlighted)

Line 102 - which is fine but should be preceded by a comma.

Corrected (highlighted)

Line 117 - led not lead?

Corrected (line 119)

Line 120 and elsewhere - However should be preceded by a semicolon.

Corrected (highlighted)

Referee #1:

1. The title of 'the role of maternal nutrition....' is a little misleading when the paper focuses almost entirely on folate and choline. It might be better to be explicit and say the role of folate, choline and one carbon metabolism in....

This has been altered as suggested to 'The role of maternal choline, folate, and one-carbon metabolism in mediating the impacts of prenatal alcohol exposure on placental and fetal development' (Line 1).

2. Some mention of the metabolic effects of alcohol that may alter other elements of nutrition essential for feto-placental development would also be useful eg what effect does alcohol have on food intake, glucose and amino acid levels, other hormones and factors known to alter placental function. Perhaps as a table.

The impact of PAE on intake and absorption of nutrients more broadly, has been mentioned in the introduction with reference to relevant studies (lines 108 – 110). An additional reference has been added here (Line 110, Kloss *et al*, 2022), to direct readers to a detailed description of the impact of PAE on a range of nutrients, with implications for feto-placental development.

While we appreciate that PAE has other impacts beyond nutrition (oxidative stress, inflammation, maternal HPA axis etc), we feel that this is outside the scope of the current review. We wanted to focus on the mechanism through which PAE may alter DNAm and gene expression and highlight evidence supporting an opportunity for nutritional intervention for PAE.

3. The paper would benefit from some discussion of placental folate and choline transport and the cellular mechanisms involved.

Mechanisms of placental folate transport have been added at lines 125 - 133, and placental choline transport at lines 151 – 158.

4. Graphical abstract and Figure 1. Identify that the circles on the DNA molecule represent methylation.

This has been added in the figure legends (Line 46 and line 184)

5. Figures 1, 2 and 3 legends should include the references from which the data has been drawn.

References have been added to the figure legends (Lines 188 - 189, 354 – 366, 469 – 471)

6. Figure 2B Should be titled 'Alcohol exposure and folate deficiency' otherwise it could be read as alcohol deficiency

This has been altered as suggested, figure 2, panel B.

7. The conclusions recommend a minimum choline dose but makes no mention of the need for a greater dose of folate during pregnancy. Should this be included?

A higher dose of folic acid would not currently be recommended in an uncomplicated pregnancy as there are concerns that the high levels of folic acid in some prenatal supplements (e.g. Elevit contains 800ug) together with fortification of foods is already resulting in high levels of unmetabolized folic acid in some pregnant women, which may have detrimental effects on fetal neurodevelopment. Rather, it is suggested to emphasise the importance of achieving the **recommended** intake. Choline on the other hand is not usually added to prenatal vitamin supplements and is not fortified in foodstuffs. Very few pregnant women meet the adequate intake level of 450mg from dietary sources and this amount is thought to underestimate requirements during pregnancy. In addition, while alcohol consumption has been demonstrated to reduce absorption of folate, there is currently no evidence that choline absorption is directly impacted. Given that up to 80% of women consume alcohol before realising they are pregnant, choline supplementation would be a safer and more effective option for supplying additional methyl groups than potentially over supplementing with folate.

Referee #2:

The review makes a nice case for the effects of alcohol mediated through, and exacerbated by, reduced nutrients and effects on methylation. It is clear that there are different mechanisms **depending on the time in pregnancy of exposure to alcohol**. I think it would be helpful to include a figure that shows the effects of alcohol/ reduced choline and folate over the course of pregnancy, ie early effects on blastocyst, later effects on placentation and early brain development, and even later effects on placental vascularity, "activity" (transporters etc) and fetal brain development/maturation.

After careful consideration, the authors felt that the complexities of timing, dose and duration of folate/choline intake and PAE in the available studies could not be clearly and accurately represented in a figure. For example, many of the animal studies that examine outcomes following PAE and choline/folate deficiency or supplementation, use models of chronic PAE throughout gestation, or for the majority of gestation. There are currently insufficient studies examining the effects of PAE at specific stages of pregnancy on different backgrounds of dietary folate/choline, to be able to determine outcomes at particular stages of placenta and fetal brain development. In addition, fetal brain development at the end of pregnancy in rodents is approximately equivalent to the end of the second trimester in humans. We have not included rodent studies of the impact of alcohol/folate/choline on the brain in the neonatal period (third trimester equivalent) as this does not include effects on placental function as a contributor to brain development.

Specific comments:

Page 6-7: Is there any information on folate or choline uptake into the blastocyst or early embryo in vivo (ie prior to placentation?)

As far as we are aware, folate and choline content of *in vivo* developed blastocyst/embryo has only been examined following *in vitro* culture with addition of labelled molecules to the culture medium. This information has been added in lines 125-130 and lines 146 – 151.

Line 185-186: Is there a reference for 'This may be particularly important during early pregnancy as the developing blastocyst is highly sensitive to changes in micronutrient availability.'

References have been added (lines 206 – 207).

Lines 248-249: The mechanism for the link between reduced availability of methyl groups and poor placentation is not discussed. What is the hypothesized mechanism? Is this due to reduced proliferative capacity or is it a specific epigenetic effect secondary to hypomethylation? Does folate also affect igf2 levels?

Reduced trophoblast viability, proliferation and expression of genes in pathways regulating invasion, have been hypothesised to underlie poor placentation. Further clarification of this has been added in lines 266 – 268 and lines 272 – 274.

Whilst there is evidence that Igf2 plays a role in regulating placentation (*Kent et al, Akt1 and insulin-like growth factor 2 (Igf2) regulate placentation and fetal/postnatal development. Int J Dev Biol. 2012;56(4):255-61*), there is currently no evidence that folate impacts expression of Igf2 **in the placenta**.

Concluding paragraph: makes a case for raising awareness about choline and folate intake in the periconception period, but the review also makes a case for the importance of both of these throughout pregnancy, and has evidence for an interaction of alcohol with either folate or choline deficiency in all stages of pregnancy.

This has been clarified in lines 480 – 487, lines 490 – 491, and line 495.

Dear Professor Moritz,

Re: JP-SR-2023-283556R1 "The role of maternal choline, folate and one-carbon metabolism in mediating the impact of prenatal alcohol exposure on placental and fetal development" by Sarah E Steane
James S M Cuffe
Karen M Moritz

I am pleased to tell you that your Symposium Review article has been accepted for publication in The Journal of Physiology, subject to any modifications to the text that may be required by the Journal Office to conform to House rules.

NEW POLICY: In order to improve the transparency of its peer review process The Journal of Physiology publishes online, as supporting information, the peer review history of all articles accepted for publication. Readers will have access to decision letters, including all Editors' comments and referee reports, for each version of the manuscript and any author responses to peer review comments. Referees can decide whether or not they wish to be named on the peer review history document.

The last Word version of the paper submitted will be used by the Production Editors to prepare your proof. When this is ready, you will receive an email containing a link to Wiley's Online Proofing System. The proof should be checked and corrected as quickly as possible.

All queries at proof stage should be sent to tjp@wiley.com.

The accepted version of the manuscript is the version that will be published online until the copy edited and typeset version is available. Authors should note that it is too late at this point to offer corrections prior to proofing. Major corrections at proof stage, such as changes to figures, will be referred to the Reviewing Editor for approval before they can be incorporated. Only minor changes, such as to style and consistency, should be made a proof stage. Changes that need to be made after proof stage will usually require a formal correction notice.

Are you on Twitter? Once your paper is online, why not share your achievement with your followers. Please tag The Journal (@jphysiol) in any tweets and we will share your accepted paper with our 22,000+ followers!

Yours sincerely,

Professor Laura Bennet
Senior Editor
The Journal of Physiology
<https://jp.msubmit.net>
<http://jp.physoc.org>
The Physiological Society
Hodgkin Huxley House
30 Farringdon Lane
London, EC1R 3AW
UK
<http://www.physoc.org>
<http://journals.physoc.org>

EDITOR COMMENTS:

Reviewing Editor:

Thank you for revising the paper.

REFeree COMMENTS:

Referee #1:

I am happy with the alterations and explanations that the authors have made in response to my comments.

Referee #2:

The authors have addressed my comments.

*** IMPORTANT NOTICE ABOUT OPEN ACCESS ***

To assist authors whose funding agencies mandate public access to published research findings sooner than 12 months after publication, The Journal of Physiology allows authors to pay an open access (OA) fee to have their papers made freely available immediately on publication.

You will receive an email from Wiley with details on how to register or log-in to Wiley Authors Services where you will be able to place an OnlineOpen order.

You can check if you funder or institution has a Wiley Open Access Account here: <https://authorservices.wiley.com/author-resources/Journal-Authors/licensing-and-open-access/open-access/author-compliance-tool.html>.

Your article will be made Open Access upon publication, or as soon as payment is received.

If you wish to put your paper on an OA website such as PMC or UKPMC or your institutional repository within 12 months of publication you must pay the open access fee, which covers the cost of publication.

OnlineOpen articles are deposited in PubMed Central (PMC) and PMC mirror sites. Authors of OnlineOpen articles are permitted to post the final, published PDF of their article on a website, institutional repository, or other free public server, immediately on publication.

Note to NIH-funded authors: The Journal of Physiology is published on PMC 12 months after publication, NIH-funded authors DO NOT NEED to pay to publish and DO NOT NEED to post their accepted papers on PMC.

1st Confidential Review

16-Jan-2023